# Just around the Corner: Advances in the Optimization of Yeasts and Filamentous Fungi for Lactic Acid Production

**DOI:** 10.3390/jof10030207

**Published:** 2024-03-09

**Authors:** Nadielle Tamires Moreira Melo, Ana Caroline de Oliveira Junqueira, Letícia Ferreira Lima, Kamila Botelho Sampaio de Oliveira, Micaela Cristiane Gomes dos Reis, Octávio Luiz Franco, Hugo Costa Paes

**Affiliations:** 1Postgraduate Program in Genomic Sciences and Biotechnology, Catholic University of Brasília (UCB), Brasília 71966-700, Brazil; nadytamires@gmail.com (N.T.M.M.); lethicialima55@gmail.com (L.F.L.); kamilasampaio18@gmail.com (K.B.S.d.O.); micaelacristiane.reis@gmail.com (M.C.G.d.R.); ocfranco@gmail.com (O.L.F.); 2Department of Molecular Biology, University of Brasília, Brasília 70790-900, Brazil; anacarolineoj@gmail.com; 3Center for Proteomic Analysis and Biochemistry (CAPB), Catholic University of Brasília (UCB), Brasília 71966-700, Brazil; 4Postgraduate Program in Molecular Pathology, University of Brasília (UnB), Brasília 70910-900, Brazil; 5S-Inova Biotech, Postgraduate Program in Biotechnology, Catholic University Dom Bosco (UCDB), Campo Grande 79117-900, Brazil; 6Clinical Medicine Division, Medical School, University of Brasília, Brasília 70910-900, Brazil

**Keywords:** lactic acid, yeast, carbon source, metabolic engineering

## Abstract

Lactic acid (LA) production has seen significant progress over the past ten years. LA has seen increased economic importance due to its broadening use in different sectors such as the food, medicine, polymer, cosmetic, and pharmaceutical industries. LA production bioprocesses using microorganisms are economically viable compared to chemical synthesis and can benefit from metabolic engineering for improved productivity, purity, and yield. Strategies to optimize LA productivity in microorganisms on the strain improvement end include modifying metabolic routes, adding gene coding for lactate transporters, inducing tolerance to organic acids, and choosing cheaper carbon sources as fuel. Many of the recent advances in this regard have involved the metabolic engineering of yeasts and filamentous fungi to produce LA due to their versatility in fuel choice and tolerance of industrial-scale culture conditions such as pH and temperature. This review aims to compile and discuss metabolic engineering innovations in LA production in yeasts and filamentous fungi over the 2013–2023 period, and present future directions of research in this area, thus bringing researchers in the field up to date with recent advances.

## 1. Introduction

Lactic acid (LA), or 2-hydroxypropanoic acid (C_3_H_6_O_3_), is a compound the molecular structure of which contains a carboxyl as terminal group and a hydroxyl group in position 2, the chirality of which yields two enantiomers, L- and D-lactate. LA can be readily converted into several useful chemicals such as pyruvic and acrylic acids, 1,2-propanediol, and lactate ester [1]. Some physicochemical properties of LA, such as its slight acidity (pK_a_ = 3.86), its carboxyl and alcohol groups, its biodegradability, non-toxicity, hygroscopicity, and chemical and thermal stability make it useful in the food, cosmetic, polymer, and pharmaceutical industries [2]. Due to its pK_a_, it is mostly found in ionized, lactate form in neutral milieus, which matters when considering its toxicity in high concentrations, as only the undissociated form can diffuse through biological membranes. In organisms that make them, the L- and D- isomers are made from pyruvate in one step by the action of the enzyme lactate dehydrogenase (LDH), which also has L- (EC number 1.1.1.27) and D- (EC 1.1.1.28) forms according to the product.

LA has GRAS (Generally Regarded As Safe) status and is recognized as such by the US Food and Drug Administration, thus being certified as safe for all food applications. It has been in use in food for almost 120 years [3,4,5]. The global international market for LA in 2022 was valued at around 3.1 billion dollars. The estimate is that it should grow at a compound annual growth rate (CAGR) of 8.0% from 2023 to 2030 [6]. As a monomer, LA is the building block of polylactide (PLA), a biopolymer that has become an important input for the bio-based industry since it is compostable, biocompatible, and a possible substitute for conventional plastics of petrochemical origin that can be produced from renewable sources in previously established bioprocesses [7].

Fermentation to produce LA is among the oldest industrial bioprocesses, first discovered in 1780 by CW Scheele, who initially considered it a component of milk. Later, in 1789, it was given its name by Lavoisier [3]. It began to be produced by fermentation in the United States in 1881 and in Europe in 1895 [8]. Production through biological routes using naturally producing or genetically manipulated microorganisms makes LA economically viable for many applications, including PLA production [4]. Fermentation using engineered strains tends to be more advantageous, as microorganisms can be modified to produce only one of the stereoisomers, which generally results in an enantiomerically pure product, desirable for polymerization. This facilitates the process of purifying and processing the monomer. There is greater industry interest in L-LA production, since metabolic conversion in humans is much faster compared to D-LA. Therefore, the levorotatory form is preferable in the food sector and in medicine [8,9,10,11]. Production by chemical synthesis, in turn, generates a racemic mixture and has a high production and energy cost [12]. It must be emphasized, however, that the mixing of different proportions of the two enantiomers influences the physicochemical properties of the polymer, which may broaden its range of applications [13], though that still requires the enantiomers to be produced separately and then mixed according to desired proportions.

Currently, the main means for producing LA is through fermentation using microorganisms such as bacteria, filamentous fungi, and yeast [1]. Bacteria that naturally synthesize it are called lactic acid bacteria (LAB), such as the genera *Carnobacterium*, *Enterococcus*, *Lactobacillus*, *Lactococcus*, *Leuconostoc*, *Pediococcus*, *Streptococcus*, *Tetragenococcus*, *Vagococcus*, and some members of the *Bacillus* genus. Some filamentous fungi are also natural producers, such as several members of the genera *Mucor* and *Rhizopus* [14].

When it comes to LA-producing microorganisms, their morphological, physiological, and biochemical characteristics must be considered to establish a bioprocess according to the carbon source used, resistance to more acidic pH, nutrients, and production time [1,7,15]. Currently, around 90% of lactic acid on the global market is produced by LAB. The most used are lactobacilli: *L. helveticus*, *L. lactis*, and *Lactiplantibacillus plantarum*, all natural producers. Several strains of bacteria that produce lactic acid have been genetically engineered to increase production and improve stereospecific purification [14,16].

Bacteria generally produce lactic acid in the form of a racemic mixture, since they have the enzymes L-lactate dehydrogenase (L-LDH) and D-lactate dehydrogenase (D-LDH), which interferes with the PLA synthesis process. Several research groups have already carried out the overexpression of the L-LDH gene as well as the deletion of the D-LDH gene with the aim of creating strains that generate a stereospecific product at increased amounts [15,17,18]. Most LAB do not grow below pH 4; although the pKa of lactic acid is 3.86, there is a fragility on the part of the bacteria in the presence of acids in the culture medium. This harms metabolism and, consequently, bacterial survival. Furthermore, the lack of nutrient supplementation makes it difficult to produce and purify lactic acid [19].

Although the production of LA at an industrial scale is at present normally carried out by LAB, new production technologies have been developed through genetic modifications using yeast. In addition to being more robust microorganisms, yeasts are resistant to industrial conditions, such as fluctuations in pH, pressure, and temperature; they also tolerate minimal media and low-cost carbon sources to produce LA. These advantages can yield greater economic viability to the process with yeast as the production platform [20,21].

Even though it poses challenges in terms of metabolic engineering and synthetic biology strategies, advances have been made to optimize microorganisms for LA production by modifying metabolic routes to avoid product degradation, the formation of more than one isomer, or unwanted coproducts; to optimize LA export across the cell membrane, among other possibilities to achieve high yield and productivity [22]. Examples of strategies that seek to accomplish these goals include (i) the introduction and comparison of LDHs from various source organisms [23], (ii) the deletion of genes that encode different pyruvate decarboxylases (Pdc’s) [24], (iii) the screening and selection of spontaneous mutants that are tolerant to LA [25], (iv) the overexpression of genes for LA transporters [26], (v) the overexpression of hexose transporters for improved glucose uptake [27], (vi) the deletion of genes encoding enzymes that compete with LDH for cytosolic NADH [28], (vii) the targeted engineering of lactate tolerance (reviewed in [29]), and (viii) the co-expression of more than one heterologous LDH [30].

While LA-related bioprocesses are among the best studied for small organic acids [31], further improvements will be needed if PLA is to become competitive against petrochemical plastics; to this end, yield will have to be maximized, productivity will have to reach at least 100 g/L, the lactate will have to be at least 99% pure, and the strain will have to be tolerant to impurity-rich and low-cost carbon sources. These requirements remain as important today as when a landmark review of the first 16 years of yeast engineering for lactate production listed them more than 13 years ago [19].

This review will compile and discuss the last decade (the period from 2013 to 2023) of results of attempts to generate and improve lactate-producing strains deploying the aforementioned strategies, with a focus on industrial yeasts, but also showing results in some filamentous fungi. We expect the present review to be useful to readers interested in the metabolic engineering end of the LA production chain, and in using yeasts as a model. Thus, we mainly focus on genetic strategies, with a secondary focus on related aspects such as acidity tolerance and carbon source choice, which must be considered when developing engineering strategies. For a discussion on bioprocess challenges in LA production in industrial settings, and considering both fungi and bacteria, we suggest the recent review by Huang et al. [32]. For a broader review on LA-producing microorganisms and carbon sources used in industrial settings, the reader may consider the review by Abedi and Hashemi [33].

## 2. Recent Metabolic Engineering Approaches to Improve Lactic Acid Production in Yeast

Yeasts need to be engineered produce LA. This can be achieved through a combination of the heterologous overexpression of L- or D-LDH genes and the deletion of genes involved in the formation of unwanted byproducts [34,35]. Numerous yeasts, including *Saccharomyces cerevisiae*, *Zygosaccharomyces bailii* [36], *Torulaspora delbrueckii* [37], members of the *Candida* genus [38], *Komagataella phaffii* [22,39,40,41], *Scheffersomyces stipitis* [42], and others, have been engineered, but in no case were viable yields for industrial production achieved [19,43]. The classic example of a yeast engineered to produce LA is *S. cerevisiae*, a so-called conventional yeast that is easy to manipulate [20,21]. However, the yield of LA obtained by most strains is low when compared to the yield obtained by LAB. [14,44].

There are many biotechnological challenges to produce LA, mainly regarding the cost–benefit of the process. Reducing the production cost of L-lactic acid monomer is one of the limiting factors. The cost must be below 0.8 EUR/kg for the process to be economically viable [45]. Thus, metabolic engineering is a powerful biotechnological tool to improve LA production parameters in different microorganisms, seeking a higher purity, acid tolerance, the use of a cheap carbon source, and well-established industrial parameters [46]. The first construction of a yeast strain for LA production was described almost 30 years ago, resulting in the production of 12 g/L in an *S. cerevisiae* strain modified by the introduction of a multi-copy vector of the LDH gene from *Lacticaseibacillus casei* [47]. Since then, numerous genetic strategies have been evaluated for improving LA production in this and other genera of yeast and filamentous fungi [20,21,22].

The yield and productivity of LA obtained by *S. cerevisiae* and other yeasts engineered solely with the introduction of an LDH gene are low relative to bacteria. This is due to the strong tendency in these yeasts to channel pyruvate elsewhere. In Crabtree-negative yeasts grown aerobically, it goes into biomass production. In Crabtree-positive ones or in hypoxic conditions, it goes into alternative fermentation products such as ethanol. Like lactate, acetaldehyde can be produced from pyruvate in one enzymatic step, and just one extra step is necessary to produce ethanol or acetate. Depending on the carbon source, more distant products that can interfere with LA yields include arabitol and glycerol. This underlies the classical approach of reducing the flux of pyruvate through competing pathways to increase the yield of D- or L-LA: the deletion of one of the six pyruvate decarboxylase (EC 4.1.1.1) genes in *S. cerevisiae*, combined with those of both the cytosolic (EC 1.1.1.8) and mitochondrial (EC 1.1.5.3) glycerol 3-phosphate dehydrogenases (Gpd) and of the alcohol dehydrogenase (Adh; EC 1.1.1.1), is an example of this principle taken to an extreme [48]. 

Furthermore, in the development of bioprocesses, the key parameters (production, productivity, and yield) are not always connected. One must also consider that optimization for a specific purpose, such as increasing the conversion yield of a cheap substrate into a high-value-added product, can result in low productivity due to the impact of metabolic engineering on microbial growth [49].

## 3. Common Metabolic Engineering Strategies

Microbial engineering based on synthetic biology can facilitate the large-scale synthesis of target products, including LA in yeast. This section discusses examples of recent strategies that resulted in high-performing strains that get closer to commercial competitiveness, and going beyond the mere introduction of an LDH gene.

*S. cerevisiae* is the preeminent species engineered to produce LA from hexose sources. A strain with just the introduction of the gene for a bovine L-LDH had a yield of 0.16 g/g on glucose and a productivity of 0.3 g∙L^−1^∙h^−1^ of lactate. As discussed above, one of the most common engineering steps is to delete a Pdc gene [50,51], and when the researchers deleted *PDC1* in this strain, there was an increase in LA yield to approximately 0.65 g/g and in productivity to 0.77 g∙L^−1^∙h^−1^ [21]. *S. cerevisiae* has an advantage for growth on glucose due to its Crabtree-positive phenotype, and has been the preferred species for lactate production from this carbon source, as shown by the diversity of strains reported in Table 1. Challenges in engineering this strain revolve around the fact that it is difficult to eliminate acetaldehyde production completely due to the presence of six *PDC* genes, with Pdc1, Pdc5, and Pdc6 accounting for most of the enzymatic activity. Thus, further performance yields accrued from the deletion of more *PDC* genes, in combination with the expression of one or more LDH genes. The result is strains of *S. cerevisiae* capable of producing more LA with little or no ethanol formation [19,20,30,52,53]. Although these modifications did result in reduced ethanol formation, the improvement in LA production was still far from ideal. This prompted further work, which aimed to simultaneously delete the genes that encode the enzymes Pdc1 and Adh1, and introduce the gene that encodes the bovine L-LDH. The removal of Adh1 seeks to ensure that, even with residual Pdc activity from the remaining isozymes, carbon will not flow to ethanol production. The new strain performed better than the one presented in the previous work, with a yield of 0.75 g/g and a productivity of 1.14 g∙L^−1^∙h^−1^, proving that the double deletion was more efficient [44].

Several issues remained unresolved from the Pdc-/Adh1- genotype in *S. cerevisiae* for LA production from glucose. For one thing, the incomplete elimination of Pdc activity, combined with the deletion of alcohol dehydrogenase, results in the accumulation of acetaldehyde. It has been postulated that this would perturb acetyl-CoA formation and negatively affect cell growth, and a recent, ingenious solution to this hurdle was to introduce a less costly metabolic bypass to the native acetate pathway by adding the gene of a bacterial acetylating acetaldehyde dehydrogenase to the genome of the yeast, which resulted in improved yeast growth while increasing lactate production [48].

Another issue specific to hexoses as a carbon source is that cytosolic glycerol 3-phosphate dehydrogenases are quite adept at diverting carbon to glycerol synthesis [54], which has been countered with the removal of the genes coding for them, resulting in further improved strains ([48] and Table 1). 

An often-neglected aspect of lactic acid production in yeast is the redox balance in the cytosol. The conversion of glucose to lactate does not have a net consumption or production of NADH, but any pathways competing for NADH in the cytosol may deprive LDH of its essential co-substrate. A recent strategy has been to delete external NADH dehydrogenases (Nde), which are enzymes facing the intermembrane space from the inner mitochondrial membrane that oxidize NADH to NAD^+^ and deliver the electrons to the respiratory chain [30]. This study resulted in strain SP7 (Table 1), which achieved one of the highest titers of lactate in yeast so far.

In the context of SP7 and the work that resulted in it, the matter of using multiple copies of an LDH gene merits consideration. In the same work, as summarized in Table 1, the authors generated intermediate strains with progressively more copies of the L-LDH gene. However, the gains between three and five copies of the gene were marginal, which indicates that the amount of enzyme may not be the limiting factor in this bioprocess. Notice in the same table that two of the best performing strains, BK01 and NO.2-100, discussed in detail elsewhere in this review, have similar performances but the former has one and the latter, three LDH genes from different organisms.

Based on these results, recently Li and collaborators [55] extensively engineered redox processes to determine their effects in the large-scale production of LA in *S. cerevisiae*. They demonstrated a decrease in the cytosolic NADH/NAD^+^ ratio from 0.228 to 0.156 during LA production. However, they unexpectedly showed that by leaving Pdc activity in place, intensifying the acetate shunt and deleting *ADH1* led to an increase in lactate production and in the NADH/NAD^+^ ratio, the latter up to 0.337. In addition to testing the effectiveness of four different redox systems, they further showed that excessive NADH is deleterious for lactate production in this setting, and that intensifying its consumption results in more of the desired product. Of particular interest, they showed that the overexpression of Glt1, the native glutamate synthase (EC 1.4.1.13) of this yeast, significantly increased LA production and reduced the NADH/NAD^+^ ratio. However, it must be noted that their final, best performing strain, PK27, produced only 37.94 g/L of LA with a production yield of 0.66 g/g in YPD medium. It is not at all clear that the trade-off with preserving Pdc activity in contrast with more conventional strategies is rewarded by a better performance.

The issue of what the best redox balance is for LA production in yeast remains open, and is bound to be organism- and carbon-source-dependent (see below). How an increase in cytosolic NADH could be detrimental remains to be determined. It is possible that the corresponding decrease in NAD^+^, a necessary co-substrate of glycolysis, disturbs the carbon flow upstream of LA production. Shu and colleagues used *Rhizopus oryzae*, a filamentous fungus that naturally produces LA, to test this hypothesis [56]. At the same time, they addressed the role of a coproduct of glycolysis, ATP, the cytosolic concentration of which can allosterically inhibit glycolytic enzymes and thus reduce carbon flow through this pathway, also resulting in lower performance. They adopted a two-pronged approach: first, they used UV radiation to generate mutants lacking a functional respiratory chain, under the rationale that the lack of respiration will reduce ATP levels and thus derepress glycolysis; then, they performed fermentations in one such mutant with gluconic acid as a secondary carbon source in addition to glucose, under the rationale that being more oxidized, it would lower the NADH/NAD^+^ ratio, thus increasing glycolytic flow. They did show a lactate titer of 102.3 g/L in the best fermentation conditions tested, using 60 g/L of each sugar, and showing a consistent drop in both the ATP concentration and the NADH/NAD^+^ ratio, thus lending credence to the previously shown results in *S. cerevisiae*.

Another approach to increasing strain performance is to remove enzymes that consume lactate. While few fungi produce it [33], many can use it as a carbon source, both in the D and L forms. Cytochrome b2 (EC 1.1.2.3) is a mitochondrial oxidoreductase that can interconvert L-lactate and pyruvate much in the same fashion as LDHs, but using cytochrome c as a co-substrate instead of NADH [57,58]. Yeasts also produce a D-lactate dehydrogenase (EC 1.1.1.28), Dld1 being its name in *S. cerevisiae* [59]. We used the Cyb2 (GenBank accession number NP_013658.1) or Dld1 (NP_010107.1) sequences from *S. cerevisiae* as queries for BlastP searches against fungal genomes, which returned hundreds of hits covering most of the sequences’ lengths and with E-values close to zero, suggesting that these proteins, and presumably the ability to consume LA, are widespread in this kingdom. Both enzymes have been knocked out in engineering attempts to improve D- or L-LA production, with good results (Table 1 and references therein).

The combination of the aforementioned strategies in *S. cerevisiae* has resulted in some of the best performing strains in lactate production, with final titers as high as 142 g/L in strain SP1130 [60], combining Cyb2, Adh1, Pdc1, and Gpd1 knock-outs, the bacterial acetyl-CoA bypass, and two LDHs. Other strains producing more than 100 g/L with similar combination strategies can be found in Table 1.

The above highlights the great progress that has been made on producing LA from glucose in *S. cerevisiae*. However, hexoses as carbon sources have disadvantages. For one, sugar-producing crops compete with staple food crops for land resources, similarly to their use to make first-generation biofuel [61]. For another, sugar-rich residues from the biotransformation industry, such as sugarcane bagasse, may require physicochemical or enzymatic preprocessing to be used in fermentation, which also increases the final cost of LA (see our discussion on carbon sources below). Therefore, the competitiveness of PLA against hydrocarbon-derived plastics may require the use of cheaper carbon sources. To address this and other issues such as acidity tolerance (see below) and industrial bioreactor temperatures, other yeasts and filamentous fungi have been developed as LA-producing platforms.

In the search of an alternative yeast to produce LA, the Crabtree-negative and thermo-tolerant *Kluyveromyces marxianus* was engineered to make D-lactate at a lower pH without compromising its growth [62]. The sole change in the genotype was the disruption of the single Pdc gene (*PDC1*) by the introduction of a codon-optimized bacterial D-LDH (*ldhA*). The final strain, KMΔ*pdc1*::*ldhA*, did not generate appreciable levels of any undesired coproducts such as ethanol, acetate, or glycerol, which appear so often with *S. cerevisiae*. The yeast culture was carried out with an aeration rate of 1.5 vvm at pH 5 and when maintained at 30 °C, it reached a D-lactate titer of 42.97 ± 0.48 g/L, corresponding to a yield of 0.85 ± 0.01 g/g of glucose, a productivity 0.90 ± 0.01 g∙L^−1^∙h^−1^, and a glucose consumption rate of 1.06 ± 0.00 g∙L^−1^∙h^−1^. At 42 °C, the titer was 52.29 ± 0.68 g/L; the productivity, 1.38 ± 0.05 g∙L^−1^∙h^−1^; and the glucose consumption rate, 1.22 ± 0.00 g∙L^−1^∙h^−1^, all higher. To prove its performance in a setting closer to industry, the authors showed that using the strain on sugarcane molasses, a low-value carbon source, it reached a titer 66.26 ± 0.81 g/L and a productivity of 0.91 ± 0.01 g/g without any additional nutrients. This study demonstrated that some yeasts may be able to reach yields of interest for industrial applications with much fewer changes in their genotype than *S. cerevisiae* [62].

**Table 1 jof-10-00207-t001:** Metabolically engineered strains of yeasts and filamentous fungi used for lactic acid production, 2013–2023. Genotypes, bioprocess efficiency parameters, buffering conditions, and carbon sources.

Isomer (L, D)	Host	LDH	Complementary Modifications	Buffering Conditions	Titer g/L	Productivity (g∙L^−1^∙h^−1^)	Yield (g/g)	Carbon Source	Ref.
L	*Aspergillus oryzae*	*Bos taurus*	Disruption of native LDH gene	3% CaCO_3_ pH 6.0	30			100 g/L of starch (dextrin or maltose)	[63]
L	*A. oryzae*	*B. taurus*		3% CaCO_3_	50.1	0.29	0.51	100 g/L glucose	[64]
L	*A. oryzae*	*B. taurus*	∆*pdcA*		56.4	0.33	0.58		
L	*A. oryzae*	*B. taurus*	∆*mpcA*		65.4	0.45	0.67		
L	*A. oryzae*	*B. taurus*	∆*pcdA/mpcA*		81.2	0.67	0.81		
L	*A. oryzae*	*Lactococcus lactis*	∆*pcdA/mpcA* without BtLDH		90.1		0.91		
L	*Aspergillus niger*	*Mus musculus* (11 copies)		Non-neutralized medium	7.7			60 g/L glucose	[65]
L	*Candida glycerinogenes*	*Rhizopus oryzae*	LDH gene from *R. oryzae* was expressed under pH-inducible promoter (*PCggmt1*)	pH 5.5	3.9			100 g/L glucose	[38]
				pH 2.5	12.3				
L	*Candida sonorensis*	*Rhizopus oryzae*	*pdc1*Δ*::RoLDH pdc2*Δ	CaCO_3_	78		0.81	100 g/L glucose	[43]
L	*C. sonorensis*	*Bacillus megaterium*	*pdc1*Δ*::BmLDH pdc2*Δ		84		0.85		
L	*C. sonorensis*	*Lactobacillus helveticus*	*pdc1*Δ*::LhLDH pdc2*Δ		92		0.94		
L	*Kluyveromyces marxianus* KM5	*Lactobacillus acidophilus* and *Staphylococcus epidermidis*		Non-neutralized medium	16			50 g/L glucose	[28]
				3.5% CaCO_3_	24		0.48		
L	*K. marxianus* KM5	*L. acidophilus* and *B. taurus*		Non-neutralized medium	14.8				
				3.5% CaCO_3_	21.2				
L	*K. marxianus* BY25571	*Lactiplantibacillus plantarum*	∆*pdc1*	Non-neutralized medium	10.5		0.65	100 g/L glucose	[66]
D	*K. marxianus* BY25571	*L. plantarum*	∆*pdc1*		8.9		0.66		
L	*K. marxianus* BY25571	*L. plantarum*	∆*pdc1*	3% CaCO_3_	46.3		0.80		
D	*K. marxianus* BY25571	*L. plantarum*	∆*pdc1*		40.0		0.78		
L	*K. marxianus* BY25571	*L. plantarum*	∆*pdc1* and ∆*cyb2*	NaOH pH 6	130	2	0.98	230 g/L Jerusalem artichoke	
D	*K. marxianus* BY25571	*L. plantarum*	∆*pdc1* and ∆*dld1*		122	2	0.95		
L	*K. marxianus* YKX001	*B. megaterium*		NaOH pH 5.5	47.37	0.99	0.5	80 g/L glucose and 20 g/L xylose	[9]
L	*K. marxianus* YKX001	*Plasmodium falciparum*			50	1.04	0.55		
L	*K. marxianus* YKX001	*P. falciparum* and *B. megaterium*	Expression of Jen1 from *S. cerevisiae*, overexpression of native *PFK*, and ∆*dld1*		103	1.44		180 g/L corncob residue	
L	*Komagataellla phaffii* GLJ	*B. taurus*	Expression of Jen1 from *S. cerevisiae*	NH_4_OH pH 5	20	0.41	0.47	40 g/L glycerol	[22]
L	*K. phaffii* GLS	*B. taurus*	Overexpression of native Jen1		~28	0.67	0.67	40 g/L glycerol	
L	*K. phaffii* GLp	*B. taurus*	∆*pdc1*		30	0.15	0.65	80 g/L glycerol	
L	*K. phaffii* GLpard	*B. taurus*	∆*pdc1* and ∆*ardh*		30		0.85	60 g/L glycerol	[39]
L	*K. phaffii* GLpm	*B. taurus*	∆*pdc1* and ∆*mpc1*		10.25	0.15	0.27	40 g/L glycerol	[40]
D	*K. phaffii*	*Leuconostoc mesenteroides* (4 copies)			3.48	0.04	0.22	Methanol	
L	*K. phaffii*	*L. plantarum*	Parental strain harbors peroxisomal a CO_2_-fixation pathway. ∆*cyb2*	2M NaOH	0.2		0.85 mg/g/h	CO_2_	[67]
L	*O. polymorpha* NCYC495 *leu1.1*	*L. helveticus*	P*_MOX_*-driven LDH expression, nitrogen source optimization, and adaptive evolution		3.8	0.03	0.08	Methanol	[68]
D	*Pichia kudriavzevii* NG7	*L. plantarum*	∆*pdc1* and adaptive evolution (6% LA)	pH 3.6	135	3.66	0.75	100 g/L glucose	[69]
				pH 4.7	154	4.16	0.72		
L	*P. kudriavzevii* E1	*Weizmannia coagulans 2–6* and *B. taurus*	∆*pdc1* and ∆*dld*	Non-neutralized medium	74.57		0.93	Glucose	[70]
L	*Saccharomyces cerevisiae* SP4	*Pediococcus sinensis* (3 copies)	∆*pdc1*, ∆*cyb2*, ∆*gpd1*, ∆*nde1*		26.6		0.34	80 g/L glucose	[30]
L	*S. cerevisiae* SP5	*P. sinensis* (4 copies)	∆*pdc1*, ∆*cyb2*, ∆*gpd1*, ∆*trp1*, ∆*nde1*		35.8		0.46		
L	*S. cerevisiae* SP6	*P. sinensis* (4 copies)	∆*pdc1*, ∆*cyb2*, ∆*gpd1*, ∆*trp1*, ∆*nde1/nde2*		36.4		0.46		
L	*S. cerevisiae* SP7	*P. sinensis* (5 copies)	∆*pdc1*, ∆*cyb2*, ∆*gpd1*, ∆*trp1*, ∆*nde1/nde2*		37.8		0.48		
				Ca(OH)_2_ pH 3.5	117		0.58	Fed-batch glucose	
L	*S. cerevisiae* EJ4L	*R. oryzae*	*cdt-1*, *gh1-1*, *XYL1*, *XYL2*, *XYL3*; ∆*ald6*, ∆*pho13*	NaOH pH 6	83	0.42	0.66	10 g/L glucose 40 g/L xylose 80 g/L cellobiose	[71]
				35 g/L CaCO_3_	23.77	0.58	0.17	41 g/L lactose	[72]
L	*S. cerevisiae* SP1130	*B. taurus* and *P. sinensis japonica*	∆*pdc1*, ∆*cyb2*, ∆*gpd1*, ∆*adh1* Expression of *mhpF* and *eutE* from *E. coli*	Ca(OH)_2_ pH 4.7	142	3.55	0.89	Fed-batch glucose	[60]
D	*S. cerevisiae* JHY5330	*L. mesenteroides* subsp*. Mesenteroides*	∆*pdc1*, ∆*adh1*, ∆*gpd1/2*, ∆*dld1*, ∆*jen1.* Overexpression of *HAA1*	Non-neutralized medium	48.9	0.41	0.79	70 g/L glucose	[73]
				CaCO_3_	112	2.20	0.80	Fed-batch glucose	
D	*S. cerevisiae* JHY5730	*L. mesenteroides*	∆*adh1-5*, ∆*gpd1/2*, ∆*dld1*, ∆*pdc1*, and adaptative evolution (4% LA).	NaOH pH 3.5	82.6	1.50	0.83	Fed-batch glucose	[17]
L	*S. cerevisiae* IBB14LA1_5	*P. falciparum*	Integration of *XR*, *XDH* and *XK* genes and ∆*pdc1.*	Non-neutralized medium	2.6	0.04	0.18	xylose	[74]
D	*S. cerevisiae* YIP-J-C-D-A1	*Escherichia coli* (3 copies inserted in transposon *locus* Ty1)	∆*pcd1*/6, ∆*adh1*, ∆*dld1*, ∆*cyb2*, and ∆*Jen1.*	Ca(OH)_2_	80	1.10	0.60	Fed-batch glucose	[75]
D	*S. cerevisiae* YIP-I-J-C-D-A1	*E. coli* (3 copies)	YIP-J-C-D-A1 plus expression of *IoGAS1.*	Non-neutralized medium	85.3	1.20	0.71	Fed-batch glucose	[76]
D	*S. cerevisiae* YIP-A15G12		YIP-I-J-C-D-A1 ∆*adh5* ∆*gpd2* ∆*gpd1* ∆*adh3* ∆*adh4*		92.0	1.21	0.70		
L	*S. cerevisiae* SR8L	*L. acidophilus*	*XYL1*, *XYL2*, *XYL3*; ∆*ald6*, ∆*pho13*	CaCO_3_	13.4		0.67	20 g/L xylose	[77]
					11.2		0.11	Acid-treated spent coffee grounds	
L	*S. cerevisiae* BK01	*L. acidophilus*	Adaptative evolution (8% LA)	Non-neutralized medium	119		0.72	200 g/L glucose	[78]
L	*S. cerevisiae* PK27	*Lactobacillus lactis*, *Rhizopus oryzae*	Adaptative evolution (7% LA)	Non-neutralized medium	37.94	0.66	0.37	80 g/L glucose	[55]
L	*S. cerevisiae* NO.2-100	*L. casei*, *R. oryzae*, and *B. taurus*	∆*pdc1*,*5*,*6* and ∆*adh1.* Expression of *ALD* from *E. coli* and overexpression of *Jen1.* Adaptative evolution (6% LA)	50 g/L CaCO_3_	121.5	1.69	0.81	90 g/L glucose	[48]

As outlined above, *S. cerevisiae* is probably the yeast of choice for lactate production from glucose, but cheaper carbon sources may be better suited when the goal is to produce PLA. An attractive such source is glycerol: as a byproduct of the biofuel industry, it is generated in large quantities, does not require preprocessing, and its use in bioplastic synthesis would reduce the ecological footprint of biofuels even further. However, the glycerol obtained from biofuel plants is contaminated with remnants from the chemical catalysis of biofuel esterification, including small amounts of detergents and methanol [79,80]. The latter poses a problem for lactate production, because it inhibits the growth of many industrial microorganisms (reviewed in the introduction to [80].For that reason, much work has been carried out on developing methylotrophic yeasts for growth on glycerol. A further advantage of using glycerol is that one needs not worry about deleting *GPD* genes, as the microorganisms will face no dearth of glycerol for their physiological needs that might entail a loss of carbon from lactate production to that end. 

*Komagataella phaffii* is one such methylotroph that meets industrial demands for high productivity and low production costs [22]. Our group has introduced the bovine L-LDH into a strain of *K. phaffii* and, following the paradigm from *S. cerevisiae*, its only annotated Pdc was knocked out. The resulting strain, GLp (Table 1), proved the potential of this organism by reaching a yield of 0.67 g/g and a productivity of 0.15 g∙L^−1^∙h^−1^ [39]. A previous work had already shown that *K. phaffii* tolerates well raw glycerol from biofuel production [80].

One problem caused by using glycerol as a carbon source for LA production is that the pathway is not redox-balanced: the conversion of glycerol to lactate results in the net production of one NADH. In strains of *K. phaffii* without Pdc alone and with an LDH, such as GLp (Table 1 and [39]), the yeast is not capable of reestablishing the redox balance through the formation of ethanol. Furthermore, methylotrophic yeasts like *K. phaffii* and *Ogataea polymorpha* are Crabtree-negative and thus require hypoxic conditions to ferment, which means that the mitochondrion is not available to reoxidise excessive NADH. It is expected that this imbalance, with a continued consumption of glycerol, would deplete cytosolic reserves of NAD^+^ and consequently reduce the flow of glycolysis. This would cause an accumulation of dihydroxyacetone phosphate (DHAP), the direct glycolytic intermediate produced from glycerol 3-phosphate (G3P), which would then take an alternative fate: gluconeogenesis to glucose 6-phosphate (G6P). This would then enter the pentose phosphate pathway, at the end of which arabitol may be formed in yeasts by the reduction of ribulose coupled to the reoxidation of excess NADH, which would restore the redox balance of the yeast. Indeed, the GLp strain generates arabitol as the main coproduct of the lactic fermentation of glycerol. Figure 1 shows the relevant pathways in *K. phaffii*. Accordingly, it was previously reported that in *K. phaffii* the effect of reducing oxygen supply from 21 to 8% led to an increase in arabitol titer of 220%, reflecting a shift of respiratory metabolism to fermentation [81]. The deletion of the arabitol dehydrogenase gene (*ARDH*; EC 1.1.1.11) resulted in a strain with increased purity of LA due to the lack of arabitol, but without further improvements in productivity, which might indicate a limit to this strategy in the absence of further metabolic engineering [40].

An alternative solution to this problem is to try to force a Crabtree-negative organism to ferment glycerol in aerobic conditions by denying pyruvate access to the mitochondrion and thus preventing its use by the citrate cycle and the respiratory chain. This has been attempted in *Aspergillus oryzae* [64] and *K. phaffii* [40] by deleting the first subunit of the main pyruvate carrier (MpcA in the former species, Mpc1 in the latter). The results were surprising: in *A. oryzae*, the hypothesis was borne out and the mutant strain produced more LA than the parental one; but in *K. phaffii*, our group saw no gains in fermentative performance by the mutant strain, and the carbon flow seemed to go mainly to biomass. The reason for the difference is unknown at present, and we speculate that in *K. phaffii* increased pyruvate carboxylase activity may allow carbon to enter the mitochondrion as oxaloacetate, though further experiments will be needed to confirm this. Be that as it may, these observations serve as a cautionary tale about extrapolating results from one organism to the other, even in the case of extremely conserved proteins such as the Mpc subunits.

## 4. Membrane Transporters

Being a weak, small monocarboxylic acid, lactic acid is permeant to biological membranes in its undissociated state. With its pKa of 3.86, at the neutral pH of the cytosol of most eukaryotes, more than 99.9% of the conjugate pair exists as lactate, which cannot cross membranes due to its negative charge. As for the extracellular milieu, at pH 5, an acidity level common in yeast culture media, still more than 90% exists as lactate. This poses a problem to organisms that either produce or consume LA, as they need to secrete or take it up according to the prevailing conditions. Additionally, in the case of producing organisms, the issue of LA toxicity arises, as a high extracellular concentration of LA in an unbuffered medium will result in small amounts of it diffusing through the cell membrane in the undissociated state and getting trapped inside the cells as they ionize, thus lowering cytosolic pH, with deleterious consequences to cell physiology. The inhibition of LDH by cytosolic acidification may also decrease the productivity of the strain [27].

It is not the goal of this review to describe in detail all the ways whereby fungi transport lactate across membranes. For this, the reader is referred to a recent review on the mechanisms of tolerance to LA in *S. cerevisiae* by Peetermans et al. [29], which also explains in some detail the mechanisms of its toxicity. However, knowledge of these prompted researchers to explore the possibility of improving LA production by means of overexpressing monocarboxylate-proton symporters to pump LA out of the yeast cell, thus mitigating toxicity and perhaps increasing productivity. The best studied of these LA transporters is Jen1 from *Saccharomyces cerevisiae*, which mediates the transport of lactate, pyruvate, acetate, and propionate. Its overexpression improves LA production [27]. Our group has obtained similar results with overexpressing the Jen1 orthologue in *K. phaffii*, but curiously, not with overexpressing the original Jen1 from *S. cerevisiae* ([22] and Table 1), thus showing that the heterologous expression of genes may lead to surprising results even in closely related donor and recipient organisms. 

Table 1 has examples of strains with Jen1 overexpressed, and curiously, with its deletion too. The purpose of the latter was said by the authors [73] to be avoiding the reuptake and consumption of D-lactate, and was coupled with the deletion of *DLD1*. Their final strain, JHY5330, is among the best performing strains reported to date, but it is not clear how much of that is contributed by the *JEN1* deletion.

## 5. Improving Acidity Tolerance in Yeast and Its Influence on Lactic Acid Production 

As said before, under neutral conditions, lactic acid will dissociate into lactate and H^+^, leading to intracellular acidification. There are some defense mechanisms to reestablish the intracellular neutral pH value [82], but product accumulation during fermentation can be detrimental to cell growth and productivity; and in the extracellular medium, the lower pH favors the undissociated form of LA that can penetrate the membrane back to the cytosol, disturbing the internal neutral pH and inhibiting LDH. For this reason, neutralizing agents such as calcium carbonate, sodium hydroxide, calcium hydroxide, and ammonium hydroxide are used in the industry to guarantee efficient microbial fermentation. Calcium carbonate (CaCO_3_) is the most used, in various concentrations (Table 1), due to its low cost. However, its use results in calcium lactate formation in the final solution, which requires an extra purification step with sulfuric acid to release free lactic acid, increasing costs and leading to the formation of large volumes of gypsum [3,82].

Generating a robust industrial yeast strain tolerant to lower pH is essential to im-prove the production of lactic acid. However, selecting genes to be deleted or overexpressed that could result in an acid-tolerant yeast is not a simple goal because the cell response to stress conditions is complex. To date, most studies about lactic acid resistance use *S. cerevisiae* [29,83,84,85], but the mechanisms are not fully elucidated. By a genome-wide analysis, combining DNA microarrays and a functional screening using the nonessential gene knockout library, Kawahata et al. [84] reviewed multiple genes and showed that those in the cluster involved in metal metabolism and regulated by the Aft1p transcription factor were enriched 2.5-fold under acidic conditions in the presence of LA, indicating that metal metabolism is affected at low pH. In a follow-up study, Abbott et al. [83] verified a strong induction of Aft1p and Aft2p transcription factors, corroborating the connection between lactate and iron metabolism that includes iron uptake, retention, and incorporation. This study suggests that higher lactate concentrations can remove free iron through chelation. Again, Peetermans et al. [29] reviewed other major aspects of lactic acid tolerance in *S. cerevisiae*, which include amino acid metabolism and cell wall composition. It is worth noting that gains in lactic acid production have been obtained from the introduction of genes that confer general acid tolerance, but these tend to be marginal [73,76], as the toxicity of small organic acids does not derive merely from lowering the pH.

Due to the numerous genes involved in acid resistance, the other challenge is to determine whether a single or multiple gene deletions would result in a resistant and robust strain. Thus, multiplex modifications are available for a few conventional strains, such as *S. cerevisiae*, which could also guide the construction of non-conventional yeast strains. Kawahata et al. [84] showed 46 monogenic deletions that resulted in strains resistant to 5.1% lactic acid, and Suzuki et al. [85] found 94 monogenic knockouts resistant to 6% lactic acid (pH 2.6). The latter also compared single and combined disruptions of four genes, *dse2*, *scw11*, *eaf3*, and *sed1*, and showed that the highest LA resistance (7%) was found in strains with more than one disruption [85], lending further support to its being a multifactorial trait.

For this reason, researchers have been using adaptive laboratory evolution (ALE) to improve and investigate tolerance mechanisms in yeasts. In ALE experiments, a microorganism is cultured in the presence of a selective condition (e.g., temperature, pH, industrial hydrolysates) for long periods to allow for many generations of selection and the emergence of the desired trait [86].

Using a selective medium, Park et al. [69] isolated an acid-tolerant *Pichia kudriavzevii* strain from grape skin and engineered it by the addition of the D-LDH gene from *L. plantarum* and the disruption of the single Pdc gene (*∆pdc1*), resulting in the DK strain. However, in 50 mL flask fermentation, only 21.4 ± 3.4 g/L of LA were produced when cultured without neutralizing agents, while the addition of 5% of Ca(OH)_2_ yielded 112 ± 4.1 g/L of LA, indicating that the tolerance of this strain should be increased to improve LA production. The DK strain was submitted to successive subcultures in YPD with increasing concentrations of LA up to 6%, and the strain that produced the most LA was selected, resulting in the DKA strain. By deploying whole genome sequencing (WGS) to compare the DK and DKA strains, 585 mutations were found that could be connected to the genetic basis of the tolerance. After analyzing candidate genes based on conserved domains, a putative transcriptional regulator named *PAR1* was selected as a candidate for the main driver of the tolerant phenotype. In a reverse genetic engineering strategy, *PAR1* was deleted in the parental strain (DK), resulting in a marginally more acid-resistant strain that did not reach the same LA titer as the evolved DKA strain (28 and 55 g/L, respectively). In bioreactor cultivation with and without pH control, the highest titer, yield, and productivity (154 g/L, 0.72 g/g, and 4.16 g∙L^−1^∙h^−1^, respectively) were reached by DKA when the pH was kept at 4.7 using Ca(OH)_2_ [69].

Monogenic mutant library screening and ALE are two ends of a spectrum of experimental approaches. The first is genotype-driven and promptly establishes a nexus between a mutation and the desired phenotype, but it is limited in its ability to generate complex, multifactorial phenotypes. Further improvements are sometimes possible by combining multiple monogenic disruptions in a single strain, but often the results are disappointing because the individual genes may not act synergistically. Conversely, genotypes that correlate with a strong phenotype of the desired kind may be missed because single-gene screenings cannot account for emergent phenotypes that are not apparent from individual genotypical traits. This explains, for example, why the quadruple *S. cerevisiae* mutant described above [85] actually had a small decrease in LA tolerance relative to the double *Δdse2Δeaf3* mutant.

On the other hand, ALE-based strategies, focused as they are on the phenotype of interest, often yield superior results, as demonstrated in the *P. kudriavzevii* case above, because they allow for emergent behavior. Their disadvantage, as showcased by the same example, is that it is often difficult to reverse-engineer the phenotype, with the result that our understanding of the mechanism behind it does not advance much by analyzing the genotype of the evolved strain. To illustrate this, a few more examples are in good order.

We briefly discussed earlier a previous approach in *S. cerevisiae* in the context of combined engineering strategies to improve LA production [48]. We will now discuss the improvements of each step. The strain S.c-NO.2-100 was first engineered by the addition of three LDH genes, from *L. casei*, *B. taurus*, and *R. oryzae*, into the loci of three pyruvate decarboxylase genes (coding for Pdc1, Pdc5, and Pdc6) to reduce ethanol and improve lactic acid production. In addition, the *ADH1* gene was also deleted, resulting in a strain that produced 31.3 g/L of LA with a yield of 0.35 g/g. To prevent acetaldehyde accumulation in the cytosol, the *eutE* gene encoding an acetaldehyde dehydrogenase (A-ALD) from *E. coli* was inserted to enable acetaldehyde conversion to acetyl-CoA, which improved cell growth. Although it showed an improvement in LA production, the lactate yield was only 0.48 g/g, probably due to an increase in the intracellular pH of 152.2% relative to the parental strain. To restore the neutral pH in the cytosol by pumping out more LA, Jen1 was overexpressed, leading to 51.4 g/L of LA. After the genetic engineering steps, the resulting strain was submitted to twelve subcultures transferred every 48 hours with growing concentrations of LA up to 6%. The production of the evolved strain, S.c-NO.2-100, was 17.5% higher than the engineered one (S.c-NO.2), and in a 5-L bioreactor in fed-batch mode with glucose and buffering by 50 g/L of CaCO_3_, the evolved strain produced 121.5 g/L with a yield of 0.81 g/g [48].

In contrast, the LA-producing *S. cerevisiae* BK01, constructed and evolved by Jang et al. [78], reached virtually the same titer as the S.c-NO.2-100, 119 g/L in fed-batch mode, but without using any neutralizing agent during cultivation. The BK01 strain was generated by evolving a previously described xylose-consuming strain (SR8L) in several subcultures on 8% LA [82]. In a genome sequencing analysis comparing BK01 and the parental SR8L, 24 SNPs were identified. To determine the mutations linked to the acid-resistance mechanism, two SNP mutations were inserted separately or combined in SR8L, but the three resulting strains did not show any improvement in tolerance to LA. Then, the deletion of both genes, separately or combined, failed to mimic the ALE-induced phenotype. As previous studies mentioned above, the ALE strategy is suitable for generating robust strains, but lactic acid tolerance is controlled by complex mechanisms that limit the usefulness of reverse engineering strategies.

A third approach to inducing tolerance to LA is phenotype-oriented engineering. This consists in introducing genotype changes aimed at engendering a specific trait connected to LA tolerance. One example is a mutation that reduces the activity of the Fps1 aquaglyceroporin in *S. cerevisiae*, which is thought in turn to reduce membrane permeability to facilitated diffusion of neutral LA (Patent US20150368306A1, reviewed in [29]). Another is the deletion of the *SAM2* gene for S-adenosyl-methionine synthetase, also in *S. cerevisiae*, which is thought to impact the phospholipid composition of cell membrane in a way that also makes it less permeable to LA [87]. A study has been published on the phenotypical response of the acid-tolerant yeast *Zygosaccharomyces bailii* that indicates ways whereby it adapts to LA, and might supply avenues of investigation for this purpose [88].

## 6. Exploring Alternative Carbon Sources

We briefly mentioned earlier the use of glycerol as a cheaper alternative to glucose for LA production. Over the years, scientists have searched for various other relatively inexpensive carbon sources [18,89,90]. Microbes engineered for lactic fermentation are capable of using saccharose, glycerol, xylose, carbon dioxide, methanol, lignocellulosic biomass, lactose-rich whey waste, and others [22,63,68,72,91,92]. A simplified scheme of the pathways to assimilate carbon from all these sources into LA is shown in Figure 2.

Yeasts have received much attention as producers of LA precisely because of their versatility in fuel tolerance [93]. In addition to the previous example of glycerol, *K. phaffii* has been shown to consume methanol as a carbon source to produce lactic acid by Yamada et al. [50]. The strategy used resulted from the genetic integration of the LDH gene at the rDNA locus and post-transformation gene copy number expansion, resulting in strains GS115/S8/Z3 and GS115/S16/Z3, that, respectively, produced 3.48 and 3.26 g/L of D-lactic acid from methanol in a 96 h test tube fermentation. Another methylotrophic yeast, *O. polymorpha*, has been engineered to produce L-LA from methanol by the introduction of a bacterial L-LDH under the control of the methanol-inducible MOX promoter. By means of nitrogen source optimization and adaptive evolution, they reached a titer of 3.8 g/L. Although these numbers are low, the authors justify the pursuit by stating that methanol is a carbon source of interest for carbon dioxide capture associated with industrial processes, because it can be generated from the latter by a simple hydrogenation step [50,68].

Following a different avenue, Turner and colleagues engineered strains of *S. cerevisiae* to demonstrate the viability of sustainable lactic acid (LA) production from xylose [20] by adding genes of enzymes (*XYL1*, *XYL2*, and *XYL3*, coding for xylose reductase, xylitol dehydrogenase, and xylulokinase, respectively; all from *S. stipitis*) to channel its carbons into the pentose phosphate pathway and then glycolysis; deleting aldehyde dehydrogenase (*ALD6*) to block acetate production; and deleting the gene of the Pho13 phosphatase to preserve xylulose 5-phosphate levels [94]. Curiously, this strain, SR8L, did not undergo the deletion of Adh1 or any of the Pdc enzymes, and yet produced no ethanol when grown on xylose, reaching 49.1 g/L of LA at a yield of 0.69 g/g in buffered conditions. The group went on to develop a strain capable of making LA from lignocellulosic hydrolysates containing 10 g/L of glucose, 40 g/L of xylose, and 80 g/L of cellobiose by adding genes for a cellobiose/cellodextrin transporter and a beta-glucosidase (*cdt-1* and *gh-1*, respectively; both from *Neurospora crassa*) to enable the uptake and fermentation of this sugar in addition to the same set of modifications performed on SR8L. When fermenting a mixture of lignocellulosic hydrolysates, the resulting EJ4L strain produced 83 g/L of lactic acid with a yield of 0.66 g of lactic acid per gram of sugar [71], with only negligible levels of ethanol [20]. Because the transporter accepts lactose and the beta-glucosidase also has beta-galactosidase activity, the authors were also able to show the ability of EJ4L to produce LA from lactose-rich whey waste [72].

In a second study on *S. cerevisiae*, Novy and collaborators modified it for xylose consumption with a different strategy [74]. They transformed two strains of yeast, IBB14LA1 (Pdc1-/Pdc5+) and IBB14LA1_5 (Pdc1-/Pdc5-), both having the L-LDH from *Plasmodium falciparum* integrated into the *PDC1* locus, plus the genes for a pathway of xylose assimilation. They showed that the loss of both Pdc genes resulted in higher yields of LA from xylose (0.27 g/g against 0.8 g/g in the double knockout strain). The authors contrasted their IBB14LA1_5 strain with SR8L and discussed that while the preservation of the ethanol pathway may increase LA productivity, that may come at the cost of yield.

A recent study by Baumschabl and coworkers [67] modified *K. phaffii* into a synthetic autotroph through the Calvin cycle, in which the RuBisCO enzyme catalyzes the addition of CO_2_ to ribulose 5-phosphate. After the modification, the growth of the yeast on CO_2_ was tested; the autotrophic strains consumed CO_2_ as a carbon source, and the *AOX1* promoter controlled the expression of an L-LDH gene. This strain was capable of producing 150 mg/L of LA in approximately 200 hours of cultivation. While this amount is minute, it works as a proof of concept that LA could be a product of carbon fixation by engineered microorganisms.

Filamentous fungi are even more versatile in their use of carbon sources than yeasts [64], and *Rhizopus oryzae* has attracted attention due to its intrinsic ability to make lactic acid and hydrolyze lignocellular biomass [95]. Saito and colleagues showed L-LA pro-duction from wheat straw powder by a simultaneous enzymatic saccharification and fermentation using the wild-type *Rhizopus oryzae* NBRC 5378 strain, reaching 6 g/L with a yield of 0.23 g/g relative to the cellulose and hemicellulose content in wheat straw [96]. Vodnar and colleagues [97] showed that the NRRL 395 strain is capable of producing 48 g/L of L-LA from 75 g/L of crude glycerol from the biofuel industry using defined medium instead of a complex one such as YPD, one of few studies to do so. Earlier, Liu et al. had already shown up to 140 g/L of L-LA using the same strain and potato hydrolysate as the main carbon source [98]. These results make this fungus one of the most promising organisms for LA production [97].

Wakai et al. studied the potential of the amylolytic fungus *Aspergillus oryzae* to generate L-LA from several starches, dextrin, and maltose via the simple introduction of an LDH gene into its genome [63]. Their strain reached approximately 30 g/L of lactate from 100 g/L of each of these carbons, which further extended the range of possible fuels for this bioprocess and showed the possibility of combining it with native saccharification, thus eliminating the costs of enzymatic cocktails.

*K. marxianus*, previously mentioned in the context of redox and ATP balance during LA production, has also been shown to be capable of simultaneous saccharification and lactic fermentation on corn cob waste, a lignocellulosic carbon source [9]. Strain YKX071 was engineered by introducing the Jen1 transporter from *S. cerevisiae* under a strong promoter, overexpressing the native 6-phosphofructokinase to increase glycolytic flow, and deleting the native D-LDH. It produced 103 g/L of lactic acid at 42 °C from 180 g/L corn cob waste, thus showing that thermotolerant yeasts can compete with filamentous fungi in this process [9]. 

## 7. Gene Editing for Lactic Acid Production

As shown in several of the examples in this and previous sections, the number of metabolic engineering steps to enhance LA production and enable the consumption of other carbon sources may be large, and some yeasts and fungi may be rather difficult to engineer using conventional methods. Thus, CRISPR-Cas9 has already been deployed in this context. Two examples follow that illustrate its usefulness.

*Schizosaccharomyces pombe*, a non-conventional yeast with a low rate of homologous recombination, had its genome edited to introduce a D-LDH and the enzymes for the bacterial acetyl-CoA shunt described earlier; and to disrupt a native L-LDH, two Pdc genes, and an ADH, thus achieving a D-LA titer of 25.2 g/L at a yield of 0.71 g/g of glucose. By further introducing a cell-surface beta-glucosidase, the researchers generated a derivative strain capable of making 24.4 g/L of D-LA from cellobiose at a yield of 0.68 g/g [99].

A different approach to gene editing for LA production was deployed by Mitsui and colleagues in *S. cerevisiae*, in which they developed a CRISPR-driven genome shuffling method to induce rapid phenotype evolution under selective pressure [100] and used it to engineer LA tolerance and introduce extra copies of 13 genes for hexose uptake and glycolytic reactions to increase carbon flow towards D-LA production [51]. Their final strain reached 33.9 g/L of D-LA from 100 g/L glucose without neutralization, and 52.2 g/L with minimal neutralization.

## 8. Conclusions and Perspectives

In view of the recent advances discussed in this review, one might ask: have we reached the stated goal of a strain that meets the criteria for commercial competitiveness suggested almost 14 years ago? Many of the strains above may have already been deployed in the industry to produce LA, so in a sense, the answer may well be yes. But how close are we to producing lactic acid to make bioplastics at a competitive cost with petrochemical plastics? On this matter, issues remain to be addressed.

As we detail in this review, great strides have been made in the improvement of yeast strains for LA production relative to the early efforts based on merely introducing the gene of an LDH and deleting pyruvate decarboxylase. Refinements in metabolic engineering have included rebalancing the redox state of the cytosol during lactic fermentation; modulating lactate transporter expression to facilitate its secretion; suppressing competing pathways while preventing inhibition by accumulated products; adaptive evolution and targeted genetic engineering to induce tolerance to LA levels; and the introduction of transgenes for whole metabolic modules, enabling the use of alternative carbon sources such as lignocellulosic materials, lactose, xylose, several glucose polymers, and even carbon dioxide.

In some cases, such as the development of *S. cerevisiae* strains able to make LA from glucose, we have reached a point where the best strains are only marginally inferior to LAB in production, with the advantages of versatility in carbon source use and tolerance to stresses relevant to industrial production such as pH and temperature. In that sense, it seems that the goal of competing with petrochemical plastics is indeed just around the corner, as implied by the title of this review. However, two caveats need to be considered.

For one thing, comparing the strains in Table 1 may be difficult due to the lack of consistency in bioprocess conditions. Several of them have been studied in complex media such as YP plus a carbon source, and may behave differently in defined media, which matters because the latter are cheaper, have fewer impurities, and are overall preferable for industrial applications. Similarly, results obtained in flask fermentation are not directly comparable to those from bioreactor incubation. Therefore, researchers aiming to produce novel strains should consider the advantages of working with defined media and bioreactor cultivation in order to approach industrial conditions.

For another, all the best performing strains in terms of LA titer and yield had glucose as the main fuel, which also increases input costs for reasons we have discussed. In contrast, strains that use glycerol can be optimized for increased titers and are still far from the maximal theoretical yield, and there is also room for improvement on strains that use other low-cost carbon sources. Finally, while the induction of LA tolerance seems to have been well developed in *S. cerevisiae*, very little has been done to this end in other yeasts, which also suggests that further improvements are possible.

In summary, current directions implied by the advances we have reviewed are the development of non-conventional yeasts and filamentous fungi, especially methylotrophs, as LA production platforms rivalling the best *S. cerevisiae* strains; the broader deployment of adaptive evolution strategies, which should ideally be part of any strain development pipeline; the more systematic use of lactate transporters for strain improvement; and the use of lessons from *S. cerevisiae*, like rebalancing the redox state during lactic fermentation, in the engineering of non-conventional strains. 

We hope that this review provides the reader with some clarity regarding where this exciting field of research is headed, and what the next breakthroughs will be. While at first glance the goal may be just around the corner, turning that corner will require the creative application of principles we have outlined, and probably the exploration of novel strategies.

## Figures and Tables

**Figure 1 jof-10-00207-f001:**
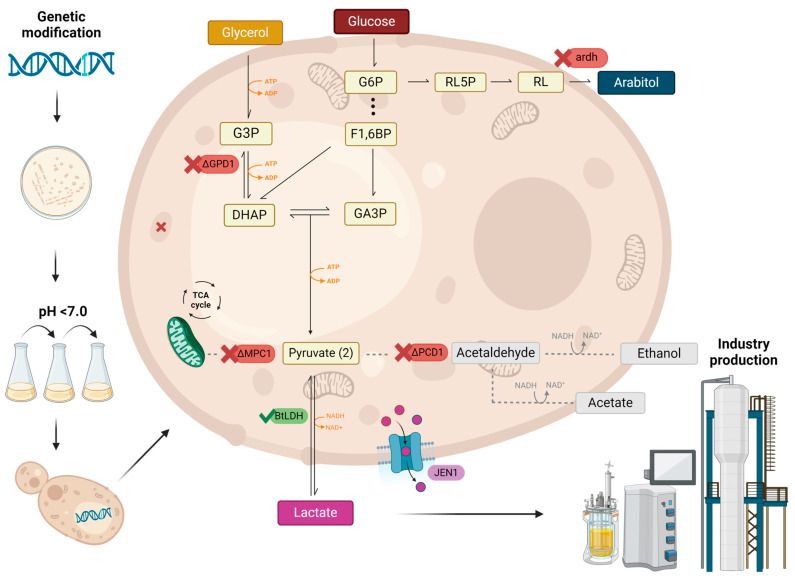
*K. phaffii* modifications to improve lactic acid production, from the selection of clones tolerant to acid to genetic engineering. This is based on the GLp strain constructed by our group by deleting (red) *PDC1* and inserting (green) the gene of the bovine L-LDH. Other changes in the yeast genotype that have been attempted include the deletion of *GPD1* and of *MPC1* [39,40,41]. See text for details. Legend: G3P, glycerol-3-phosphate; DHAP, dihydroxyacetone-phosphate; GA3P, glyceraldehyde-3-phosphate; 1,3BPG, 1,3-bisphosphoglycerate; PEP, phosphoenolpyruvate; F1,6BP, fructose-1,6-bisphosphate; ATP, adenine triphosphate; ADP, adenine diphosphate; NADH/NAD^+^, nicotinamide adenine dinucleotide; G6P, glucose 6-phosphate; RL5P, ribulose 5-phosphate; RL, ribulose; F1,6BP, fructose 1,6-bisphosphate. Made with BioRender.

**Figure 2 jof-10-00207-f002:**
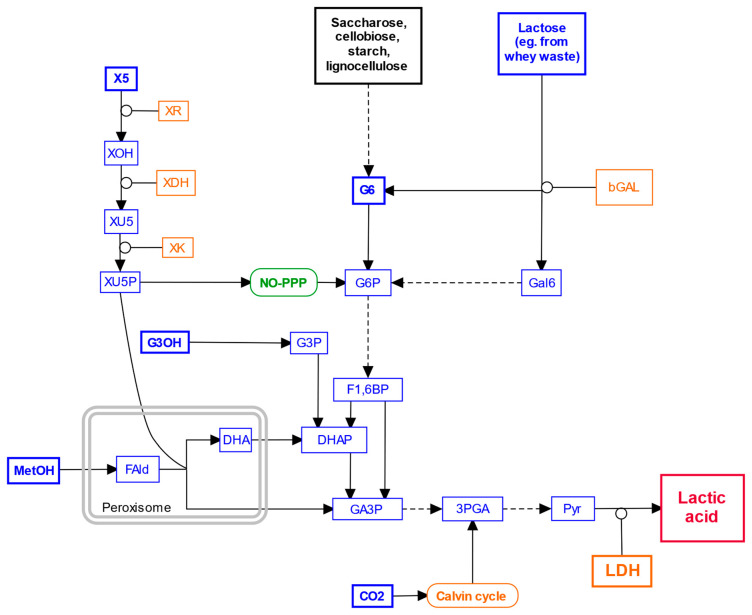
Simplified scheme of pathways leading to lactic acid from various carbon sources reported to have been used in yeast and filamentous fungi. Blue are individual metabolites, bolded for the ones fed to the target organisms in cultures; the exception is lactic acid, which is red for emphasis. Black are miscellaneous carbon sources that feed into the same pathways. Orange are enzymes and pathways that may need to be introduced as transgenes into the target organisms for them to be able to use a given carbon source; LDH is bolded for emphasis. Green is a native pathway not shown in detail for simplicity. Co-substrates, coproducts, transporters, and competing pathways are omitted for clarity. Solid arrows are single, and dashed arrows multiple enzymatic steps. **Metabolite key:** 3PGA, 3-phosphoglycerate; CO_2_, carbon dioxide; DHA, dihydroxyacetone; DHAP, dihydroxyacetone phosphate; F1,6BP, fructose 1,6-bisphosphate; FAld, formaldehyde; G3OH, glycerol; G3P, glycerol 3-phosphate; G6, glucose; G6P, glucose 6-phosphate; GA3P, glyceraldehyde 3-phosphate; Gal6, galactose; MetOH, methanol; Pyr, pyruvate; X5, xylose; XOH, xylitol; XU5, xylulose; XU5P, xylulose 5-phosphate. **Enzyme key:** bGAL, beta-galactosidase (EC 3.2.1.23); LDH, lactate dehydrogenase; XDH, xylitol dehydrogenase (EC 1.1.1.B64); XK, xylulokinase (EC 2.7.1.17); XR, D-xylose reductase (EC 1.1.1.307). NO-PPP, non-oxidative phase of the pentose phosphate pathway. See text for details. Made with PathVisio.

## Data Availability

No new data were created or analyzed in this review. Data sharing is not applicable.

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
