# Peer review of "Just around the Corner: Advances in the Optimization of Yeasts and Filamentous Fungi for Lactic Acid Production"

_jof, 2024, doi:10.3390/jof10030207_

Round 1

Reviewer 1 Report

Comments and Suggestions for Authors

The authors present a review manuscript on the production of lactic acid (LA) with a special focus on the optimization of yeasts and filamentous fungi for lactic acid production. The topic of LA production is of high interest currently and the field has been the focus of reviews in recent years. Overall, this work is systematic and comprehensive. However, some issues should be addressed by the authors. I suggest a major revision for the manuscript and the main comments are as follows:

1. Introduction can be more elaborated for microbial production of LA in industrial scale.

2. It is recommended to present the metabolic pathways for production of LA in the manuscript.  

3. Some sentences are so long that they're hard to read. They should be divided into lower-level subsections for better readability. For example,

Line 100-107: While LA- related bioprocesses are among the best-studied for small organic acids……. them more than 13 years ago.

Line 110-112: Some wild aerobic yeasts such as Kluyveromyces lactis generate LA among the by-products of carbohydrate metabolism, including ethanol and glycerol, but most need to be engineered to do it.

Line 171-173: This prompted further work, which aimed to simultaneously delete the genes that encode the enzymes Pdc1 and Adh1, which were replaced by the gene that encodes the bovine L-LDH.

4. “It is estimated that, to be economically viable, the cost of raw materials and capital investment to the LA production must be less than 55 cents per kilogram”, authors should explain how the calculations were made or cite relevant references.

5. Line 160, PDC abbreviation should be written in full as the first time it is illustrated in the text.

6. The authors are advised to check the full text properly, many places are missing literature citations. For example,

Line 243-244: While few fungi produce it, many can use it as a carbon source, both in the D and L forms.

Line 256-260: For one, sugar-producing crops compete with staple food crops for land resources. For another, ……the final cost of LA.

Line 346-349: With its pKa of 3.86, at the neutral pH of the cytosol of most …….still more than 90% exists as lactate.

7. As it is, the manuscript does not offer new insights on recent developments. This reviewer strongly recommends authors provide more information on perspective on LA production, separation and purification.

8. The figure quality should be improved. The words in the graphic are too small to read.

Comments on the Quality of English Language

 An accurate English revision is recommended.

Author Response

Dear Sir/Madam,

thank you very much for your thorough review of our manuscript. We believe your suggestions have resulted in a much improved version, which we are resubmitting for your appreciation.

We address your comments point by point below. You'll find a version of the manuscript uploaded with edits highlighted in red or blue, along with the unmarked one.

  1. Introduction can be more elaborated for microbial production of LA in industrial scale.

Our response: We have described the microbial production of LA in industrial scale as requested, starting on line 85:

            “Currently around 90% of lactic acid on the global market is produced by LAB. The most used are lactobacilli: L. helveticus, L. lactis and L. plantarum, all natural producers. Several strains of bacteria that produce lactic acid have been genetically engineered to increase production and improve stereospecific purification [14,16].

Bacteria generally produce lactic acid in the form of a racemic mixture, since they have the enzymes L-lactate dehydrogenase (L-LDH) and D-lactate dehydrogenase (D-LDH), which interferes with the PLA synthesis process. Several research groups have already carried out overexpression of the L-LDH gene as well as deletion of the D-LDH gene with the aim of creating strains that generate a stereospecific product at increased amounts [15,17,18]. Most LAB do not grow below pH 4: although the pKa of lactic acid is 3.86, there is a fragility on the part of the bacteria in the presence of acids in the culture medium. This harms metabolism and, consequently, bacterial survival. Furthermore, the lack of nutrient supplementation makes it difficult to produce and purify lactic acid [19].”

  1. It is recommended to present the metabolic pathways for production of LA in the manuscript.  

Our response: We agree and have have included Figure 2 on page 16 to summarise the pathways from various carbon sources to lactic acid.

  1. Some sentences are so long that they're hard to read. They should be divided into lower-level subsections for better readability. For example,

Line 100-107: While LA- related bioprocesses are among the best-studied for small organic acids……. them more than 13 years ago.

Our response:  We've split this sentence into two after "sources". (Lines 132-8)

Line 110-112: Some wild aerobic yeasts such as Kluyveromyces lactis generate LA among the by-products of carbohydrate metabolism, including ethanol and glycerol, but most need to be engineered to do it.

Our response: We've removed this sentence altogether. The paragraph now begins as in line 151.

Line 171-173: This prompted further work, which aimed to simultaneously delete the genes that encode the enzymes Pdc1 and Adh1, which were replaced by the gene that encodes the bovine L-LDH.

Our response: We've slightly altered this sentence to replace the subordinate clause by a coordinate one, "and introduce the gene that encodes the bovine L-LDH". (Line 246)

  1. “It is estimated that, to be economically viable, the cost of raw materials and capital investment to the LA production must be less than 55 cents per kilogram”, authors should explain how the calculations were made or cite relevant references.

Our response: We've changed the wording of the paragraph and added a more up-to-date reference. As you can read on page 3, lines 162-4:

Reducing the production cost of L-lactic acid monomer is one of the limiting factors. The cost must be below 0.8 EUR/kg for the process to be economically viable.

  1. Line 160, PDC abbreviation should be written in full as the first time it is illustrated in the text.

Our response: We've introduced (Pdc) in the previous line when the enzyme is first mentioned. (Line 232)

  1. The authors are advised to check the full text properly, many places are missing literature citations. For example,

Line 243-244: While few fungi produce it, many can use it as a carbon source, both in the D and L forms.

Our response: We have added a reference that reports that native lactic fermentation has only been reported in Rhizopus and closely related fungi (Line 345).

The contention we made about fungi being generally able to metabolise LA may have been too bold, because it hasn’t been systematically studied. What we know is that the Cyb2 and Dld1 genes have hundreds of high-similarity alignment hits in Fungi using BlastP with S. cerevisiae sequences, suggesting that the ability to metabolise LA is indeed widespread. We have added a sentence starting at line 349 to reflect this:

“We have used the Cyb2 (GenBank accession number NP_013658.1) or Dld1 (NP_010107.1) sequences from S. cerevisiae as queries for BlastP searches against fungal genomes returns hundreds of hits covering most of the sequences’ lengths and with E-values close to zero, suggesting that these proteins, and presumably the ability to consume LA, are widespread in this kingdom.”

Line 256-260: For one, sugar-producing crops compete with staple food crops for land resources. For another, ……the final cost of LA.

Our response: the analogy here was with first-generation biofuels. We’ve added a clarification and a reference (lines 363-4).

Line 346-349: With its pKa of 3.86, at the neutral pH of the cytosol of most …….still more than 90% exists as lactate.

Our response: There is no reference for this because it follows from the Henderson-Hasselbach equation. The demonstration is below.

The HHE is:

pH = pKa + log ([A-]/[HA])

where log is the decimal logarithm. Specifying terms for the Lac-/HLac pair gives us

pH = 3.86 + log ([Lac-]/[HLac])

For neutral pH:

7 = 3.86 + log ([Lac-]/[HLac]), then

log ([Lac-]/[HLac]) = 3.14, meaning that

[Lac-]/[HLac] = 10^(3.14), thus

[Lac-]/[HLac] = 1380,

meaning that at neutral pH the concentration of dissociated lactate is more than a thousand times the concentration of the undissociated form, thus the proportion of the latter, as stated in the text, is less than 0.1%.

Solving for pH 5:

5 = 3.86 + log ([Lac-]/[HLac]), then

log ([Lac-]/[HLac]) = 1.14, meaning that

[Lac-]/[HLac] = 10^(1.14), thus

[Lac-]/[HLac] = 13.8,

meaning that at pH 5 the concentration of dissociated lactate is more than ten times the concentration of the undissociated form, thus the proportion of the latter, as stated in the text, is less than 10%. This is not coincidentally a hundred times more than at pH 7, because the pH scale is a decimal logarithm one. We don't say that in the text, but each pH unit we can save on buffering means ten times less gypsum in the end, which makes even partial tolerisation of LA-producing strains a worthwhile endeavour.

  1. As it is, the manuscript does not offer new insights on recent developments. This reviewer strongly recommends authors provide more information on perspective on LA production, separation and purification.

Our response: LA production, separation and purification have been reviewed quite recently. We have extended the final paragraph (lines 142-50) of the introduction to explain that our emphasis is on genetic engineering of yeast and filamentous fungi, that our target readership are scientists interested in producing or improving fungal strains for LA production, and pointing readers to reviews on the bioprocess side of LA production for those interested.

  1. The figure quality should be improved. The words in the graphic are too small to read.

Our response: We agree and have improved the quality of the figure by increasing the font size.

Reviewer 2 Report

The authors reviewed the genetically modified yeast and fungal strains for the production of lactic acid, one of the most important bulk chemicals with application in different industries such as food, cosmetic, chemical, pharmaceutical, etc. The improvement of the fungal and yeast strains using genetic and metabolic engineering could solve several issues connected to lactic acid bacteria, which are mainly used as production strains for lactic acid production. Therefore this review could be relevant to both the academy and industry. 

General comments:

1.       The names of the genes are written in small letters in italics. The authors should check and correct gene symbols.

2.      Most of the references published after 2014 should be deleted, especially in chapters 2 and  3

3.      Instead, the term “neutral form” of LA should be LA in an “undissociated state”.

Specific comments

1.      Lines 25  "tolerance of industrial-scale culture conditions "

The authors should be  precise regarding the "industrial-scale culture conditions "(pH, T, mixing,…)

2.      Line 120 “compared to the yield obtained by bacteria”

The authors should be more precise about the microorganism e.g. LAB.

3.      Line 122-123“It is estimated that, to be economically viable, the cost of raw materials and capital investment to the LA production must be less than 55 cents per kilogram”

The sentence is unclear. Did the authors mean that production should be below  55 cents per kilogram? The authors should rewrite the sentence.

4.      The authors should add text on the main problems regarding the traditional  LA production with LAB, e.g. neutralization of broth during fermentation, demanding nutritional requirements of LAB (cost for growth medium, which can be up to 35% of the cost of production), substrate and product inhibition, problems with product isolation from culture broth,… Furthermore, the authors should explain the advantages of LA production with yeasts and fungi strains over the production of LA with LAB. Comparison of LA production's productivities/yield/economic parameters with the production of the LA with recombinant strains (yeast and filamentous fungi) would also be interesting for the Journal readers. 

5.      Table 1.

The table presents studies on LA production with metabolically engineered microorganisms in the last ten years. However, references 51, 38, 52, and 53 were published before 2014. The authors should delete those references from the table 1.

The title of the Table 1 should be rewritten. The authors should describe the data presented in a table: bioprocess efficiency parameters. Instead yeasts and fungi should be written “metabolically engineered or recombinant yeast and fungi strains”.

Instead of "Observation" should be written "Conditions" of "pH regulation/value"

6.      Lines 134-136 “This is due to the 133 strong tendency in these yeasts to channel pyruvate into either biomass production, in 134 Crabtree-negative yeasts grown aerobically; or into alternative fermentation products 135 such as ethanol in Crabtree-positive ones or in hypoxic conditions.”

The sentence is unclear and confusing for the reader. The authors should rewrite the sentence.

7.      Line 284-286: “However, the glycerol obtained from biofuel plants is contaminated with remnants from the chemical catalysis of biofuel esterification, including small amounts of detergents and methanol. The laer poses a problem for lactate production, because it inhibits growth of many industrial microorganisms.”

The sentence is unclear. Crude glycerol from biodiesel production contains glycerol and water salts as well as detergents and catalysts in traces that could inhibit the microorganisms. The authors should rewrite the sentence.

8.      Lines 500-502 “Microbes used for lactic fermentation are capable of using saccharose, glycerol, xylose, carbon dioxide, methanol, lignocellulose biomass and others, which are essential as they provide energy for biotransformation”

The sentence is unclear. Different carbon sources can be used for the growth of microorganisms. In most microorganisms that produce LA, the production of LA is growth-related. It is produced during the exponential phase to provide energy for growth as well as during the stationary phase to provide cells with energy for vital cell functions (endogenous metabolism). Therefore, the energy source is used for growth and maintenance energy, not for biotransformation. The authors should rewrite the sentence.

Carbon dioxide is an unusual carbon source for heterotrophic microorganisms (except for recombinant strains, e.g. P. pastoris metabolically engineered). Therefore, the reviewer suggests to exclude this carbon source.

Author Response

Dear Sir/Madam,

thank you very much for your thorough review of our manuscript. We believe your suggestions have resulted in a much improved version, which we are resubmitting for your appreciation.

We address your comments point by point below. You'll find a version of the manuscript uploaded with edits highlighted in red or blue, along with the unmarked one.

Abstract:

The abstract should be improved. The Abstract should include the background information and purpose on the topic being discussed in the review, followed by the aim of the review. 

Our response: we have amended sentences to make it clear that this review is about strain improvement (as opposed, for example, to bioprocess optimisation) and metabolic engineering.

Discussion:

In the discussion section, the authors should critically summarise the findings reported in this manuscript, giving the possible future direction of research on improving LA production through metabolic engineering and adaptive evolution of microbial strains. However, in the Conclusion section, the authors are focused on the carbon source used for growth and LA bioprocess efficiency parameters. Also, the authors should summarize the paper's significance to its field. 

Our response: we partially agree with the reviewer's comment.

We have significantly expanded the Discussion (pages 18 and 19) to summarise our findings and highlight what work we think is still needed to produce superior strains.

However, we would like to retain the short paragraphs on caveats comparing production on defined versus complex media (not really carbon sources) and on flask versus bioreactor cultivation because it's our belief the reader should be reminded of these, even if they are not the focus of the review. Otherwise, readers run the risk of comparing apples and oranges. All too often we see papers on LA production that show strains that perform great on YPD in flasks, but not a word on bioreactor production using defined media.

We do mention the issue of carbon sources because that has a direct bearing on the choice of genes to be knocked into the genome of the target strain, and on the overall strategy. For example, redox imbalance is much more of an issue when the fuel is glycerol than when it's glucose. Xylose and lignocellulose/starch disaccharides require the addition of extra genes and transporters. Even the choice of using a thermo-tolerant microrganism or not is impacted by the decision to use a source that requires saccharification.

General comments:

  1. The names of the genes are written in small letters in italics. The authors should check and correct gene symbols.

Our response: This nomenclature standard applies to bacterial genes only. Most fungi follow eukaryote standards, which means genes are in all-capitals, italics, and proteins are roman, first letter capitalised. The two exceptions in this paper are genes from Neurospora crassa, which are in lowercase italics, like bacterial genes. Similarly, loss-of-function mutants in eukaryotes are marked by italics, lowercase, with a delta before or after.

If you mean that sometimes we use all-capitals, italics, and sometimes Roman, first-letter capitals, this is because when we're talking about phenotypes, we refer to enzymes and proteins. When we're talking about genetic engineering steps, we refer to genes.

  1. Most of the references published after 2014 should be deleted, especially in chapters 2 and  3.

Our response:

Our review was written during the second semester of 2023, so our starting year is 2013. We've changed "last ten years" in our text to "in the 2013-2023 period", since there haven't been new papers on LA in yeast in 2024 yet.

As for the older references, we believe the newer strategies build up on the older ones. For example, we have to contextualise redox balance studies with the issue of pyruvate decarboxylase knock-outs, or adaptive evolution in the context of previously successful genotypes, so any discussion on recent advances needs to at least acknowledge the older strategies as a starting point, as the bulk of metabolic engineering strategies for LA production are cumulative, rather that parallel approaches. This is why, while only the 2013-2023 references made into Table 1, in the text itself older references are used to introduce topics and connect to recent approaches.

  1. Instead, the term “neutral form” of LA should be LA in an “undissociated state”.

Our response:

The three instances of it have been accordingly emended as suggested.

Specific comments

  1. Lines 25  "tolerance of industrial-scale culture conditions "

The authors should be precise regarding the "industrial-scale culture conditions "(pH, T, mixing,…)

Our response: we added “such as pH and temperature” as qualifiers to this sentence.

  1. Line 120 “compared to the yield obtained by bacteria”

The authors should be more precise about the microorganism e.g. LAB.

Our response: We specify that these are yields of lactic acid produced in LABS. As you can read on page 3, line 160.

  1. Line 122-123 “It is estimated that, to be economically viable, the cost of raw materials and capital investment to the LA production must be less than 55 cents per kilogram”

The sentence is unclear. Did the authors mean that production should be below 55 cents per kilogram? The authors should rewrite the sentence.

Our response: We changed the wording of the paragraph and added a more up-to-date reference. As you can read on page 3, lines 162-4:

Reducing the production cost of L-lactic acid monomer is one of the limiting factors. The cost must be below 0.8 EUR/kg for the process to be economically viable.

  1. The authors should add text on the main problems regarding the traditional  LA production with LAB, e.g. neutralization of broth during fermentation, demanding nutritional requirements of LAB (cost for growth medium, which can be up to 35% of the cost of production), substrate and product inhibition, problems with product isolation from culture broth,… Furthermore, the authors should explain the advantages of LA production with yeasts and fungi strains over the production of LA with LAB. Comparison of LA production's productivities/yield/economic parameters with the production of the LA with recombinant strains (yeast and filamentous fungi) would also be interesting for the Journal readers. 

Our response: We partially agree with the reviewer’s comment. We describe the advantages and disadvantages of LA production in LABs. However, the focus of the article is not the comparison between LABs and yeast, but new advances in metabolic engineering to improve LA production in different yeasts and some filamentous fungi. You can read the additions on page 3, lines 142-50.

  1. Table 1.

The table presents studies on LA production with metabolically engineered microorganisms in the last ten years. However, references 51, 38, 52, and 53 were published before 2014. The authors should delete those references from the table 1.

Our response: We changed the time range to 2013-2023 to reflect the end year of our literature review. References 52 and 53 have been accordingly removed. Reference 51 is from 2015.

The title of the Table 1 should be rewritten. The authors should describe the data presented in a table: bioprocess efficiency parameters. Instead yeasts and fungi should be written “metabolically engineered or recombinant yeast and fungi strains”.

Our response: We changed the title to "Metabolically engineered strains of yeasts and filamentous fungi used for lactic acid production, 2013-2023. Genotypes, bioprocess efficiency parameters, buffering conditions and carbon sources."

Instead of "Observation" should be written "Conditions" of "pH regulation/value"

Our response: For brevity, we changed it to "buffering conditions".

  1. Lines 134-136 “This is due to the 133 strong tendency in these yeasts to channel pyruvate into either biomass production, in 134 Crabtree-negative yeasts grown aerobically; or into alternative fermentation products 135 such as ethanol in Crabtree-positive ones or in hypoxic conditions.”

The sentence is unclear and confusing for the reader. The authors should rewrite the sentence.

Our response: We have edited the sentence as follows (lines 206-9).

This is due to the strong tendency in these yeasts to channel pyruvate elsewhere. In Crab-tree-negative yeasts grown aerobically, it goes into biomass production. In Crabtree-positive ones or in hypoxic conditions, it goes into alternative fermentation products such as ethanol.

  1. Line 284-286: “However, the glycerol obtained from biofuel plants is contaminated with remnants from the chemical catalysis of biofuel esterification, including small amounts of detergents and methanol. The latter poses a problem for lactate production, because it inhibits growth of many industrial microorganisms.”

The sentence is unclear. Crude glycerol from biodiesel production contains glycerol and water salts as well as detergents and catalysts in traces that could inhibit the microorganisms. The authors should rewrite the sentence.

Our response: We have added references on methanol toxicity to non-methylotrophs and on its presence in biofuel production as a catalyst. Please see lines 396-8.

  1. Lines 500-502 “Microbes used for lactic fermentation are capable of using saccharose, glycerol, xylose, carbon dioxide, methanol, lignocellulose biomass and others, which are essential as they provide energy for biotransformation”

The sentence is unclear. Different carbon sources can be used for the growth of microorganisms. In most microorganisms that produce LA, the production of LA is growth-related. It is produced during the exponential phase to provide energy for growth as well as during the stationary phase to provide cells with energy for vital cell functions (endogenous metabolism). Therefore, the energy source is used for growth and maintenance energy, not for biotransformation. The authors should rewrite the sentence.

Carbon dioxide is an unusual carbon source for heterotrophic microorganisms (except for recombinant strains, e.g. P. pastoris metabolically engineered). Therefore, the reviewer suggests to exclude this carbon source.

Our response: we agree and have removed the second part of the sentence, since it is inaccurate as you point out and not needed for the point we’re conveying. About the use of carbon dioxide, we meant to include engineered organisms, so we replaced “used” by “engineered” in the same sentence. Please check lines 623-5. For completeness, we’ve added a 2017 reference about the use of whey waste, a lactose-rich source, for LA production as well, and updated the table accordingly.

Round 2

Reviewer 2 Report

The authors revised the manuscript according to the reviewer's comments.

The authors revised the manuscript according to the reviewer's comments.